# Identifying the Bundle/Care Development Process in Clinical Risk Management: A Systematic Review

**DOI:** 10.3390/healthcare12222242

**Published:** 2024-11-11

**Authors:** Emanuele Sebastiani, Marina Scacchetti, Manuele Cesare, Massimo Maurici, Michele Tancredi Loiudice

**Affiliations:** 1Department of Biomedicine and Prevention, University of Rome Tor Vergata, 00123 Rome, Italy; 2Research and International Relations Unit, Italian National Agency for Regional Healthcare Services, 00187 Rome, Italy; 3ASL Roma 1, 00193 Rome, Italy; 4Gemelli IRCCS University Hospital Foundation, Catholic University of the Sacred Heart, 00168 Rome, Italy

**Keywords:** patient care bundle, bundle, care bundle, clinical risk, risk management

## Abstract

Background: A bundle is a set of three to five evidence-based interventions designed to improve the quality and outcomes of care processes. Numerous international studies have evaluated the effectiveness of Bundles/Care Bundles (BCB) in reducing morbidity and mortality. The Institute for Healthcare Improvement (IHI) has defined the concept of a bundle but has not outlined the development process. Objective: To identify the BCB development process in clinical risk scenarios from September 2005 to September 2023. Methods: A systematic review was conducted following PRISMA guidelines to identify studies describing the BCB development process in managing clinical risk situations. The databases consulted included PubMed, Embase, and CINAHL, along with manual searches on institutional websites. Relevant studies concerning the BCB development process were included. Results: A total of 1372 studies were retrieved, of which 16 were included. Duplicates were removed, and titles and abstracts were analyzed. The identified methods for BCB development include IHI guidelines, expert opinions, international guidelines, and flowcharts. The most common BCBs relate to the prevention of ventilator-associated pneumonia, surgical site infections, catheter-associated infections, and sepsis. Conclusions: This study has identified the development processes of Care Bundles (BCBs) in clinical risk scenarios, highlighting how these tools facilitate compliance monitoring among members of the healthcare team. The review has revealed effective methods for designing evidence-based BCBs. However, the scarcity of studies on the methodology for developing BCBs is a limitation, suggesting the need for further research. In Italy, there is a growing interest in the use of care packages. It is essential to encourage research that optimizes the effectiveness of intervention strategies.

## 1. Introduction

The Institute for Healthcare Improvement (IHI) has outlined key characteristics for the design of bundles but has not explicitly detailed the BCB development process. These characteristics focus on the implementation of all bundle components through practical actions that are easy to monitor, supported by solid scientific evidence, independent, and measurable [1].

This review aims to identify the BCB development process for managing clinical risk situations across various care settings from 2005 to September 2023.

A bundle consists of three to five evidence-based interventions, behaviors, or practices tailored to a specific patient population and care context. When applied together, these elements improve quality and outcomes more effectively than if implemented individually [2].

The safety of care, as indicated by Italian legislation (LAW 8 March 2017, n. 24), is also achieved through a set of activities aimed at preventing and managing the risks associated with the delivery of healthcare services, as well as the appropriate use of structural, technological, and organizational resources. Care Bundles can certainly play a fundamental role in prevention, as highlighted in the article concerning falls in hospital settings [3].

The bundle concept was developed in 2001 by the IHI and the Voluntary Hospital Association (VHA) as part of the Idealized Design of the Intensive Care Unit (IDICU) initiative, which involved 13 hospitals. The goal was to improve the reliability of processes in intensive care through a targeted set of interventions for a defined group of patients, resulting in significantly improved outcomes [1]. Interest in using bundles for patients in intensive care, particularly those requiring invasive ventilation or central lines, increased in the late 1990s after a study demonstrated that a protocol comprising distinct elements reduced in-hospital mortality for severe sepsis and septic shock [4]. This initial success led to the adoption of bundles as improvement tools in various medical and surgical specialties. For example, a literature review [5] highlighted interventions that positively impacted patient outcomes in intensive care units, focusing on morbidity, mortality, and costs. Bundles played a crucial role in the ‘100,000 Lives Program’ and the ‘5 Million Lives Campaign’, [5] both initiated by the IHI to enhance the quality of care and patient safety in diverse clinical scenarios.

Numerous examples exist in the literature, such as a single-center study showing a 51% reduction in the incidence of ventilator-associated pneumonia over two years after introducing a bundle aimed at reducing mechanical ventilation-related infections [6]. The sepsis bundle, part of the Surviving Sepsis campaign, is now widely used in intensive care units. However, adherence to the bundle and to the guidelines in the management of sepsis is low, and some studies have attempted to analyze the adherence to the bundle in relation to a reduction in mortality among patients with sepsis, severe sepsis, or septic shock, with results showing an association [7].

Further evaluations of the sepsis guidelines involved data from 15,022 subjects across 165 facilities in the United States, Europe, and South America between January 2005 and March 2008. Compliance with the bundle increased from 10.9% to 31.3% over two years, correlating with a reduction in hospital mortality from 37% to 30.8% (*p* = 0.001) [8].

In recent decades, the term ‘Care Bundles’ has been expanded to include care programs that encompass four fundamental concepts: involve, educate, perform, and evaluate [9]. Recently, the bundle has been defined as a tool to promote adherence to good clinical practices in various contexts, taking into account the specific characteristics of the facilities in which it is applied. Consequently, it may vary from one facility to another [10].

The bundles were not intended to be comprehensive care; rather, they were developed to test a theory—that is, when compliance is measured for a core set of accepted elements of care for a clinical process, the necessary teamwork and cooperation required will result in high levels of sustained performance [reliability] not observed when working to improve individual elements [1].

## 2. Methods

### 2.1. Study Protocol

Conducted a systematic review of studies describing the BCB development process in the management of clinical risk situations. This systematic review was registered in PROSPERO (International Prospective Register of Systematic Reviews; CRD42024498036). The review was conducted in accordance with the statement Preferred Reporting Items for Systematic Reviews and Meta-Analyses (PRISMA) [11].

### 2.2. Literature Search Strategy

The following databases were consulted: PubMed, Embase, and CINAHL. An additional manual search was performed on the following institutional sites: the Institute for Healthcare Improvement (IHI), Joint Commission International, Agency for Healthcare Research and Quality (AHRQ), World Health Organisation (WHO), and European Centre for Disease Prevention and Control (ECDC). The complete search strategy is detailed in [Appendix A]. The database searches were performed from 2005 to September 2023. The publication language of these studies was limited to English, Italian, and Spanish. A more specific search strategy was developed to identify primary studies, adapted for each database, using both Medical Object Headings (MeSH) terms and free text keywords in the title and abstract fields: “patient care bundle”; “bundle”; “care bundle”; “evidence-based care bundle”; “evidence-based care bundle”; “best practice bundle”; “care package”; “care pathway”; “care intervention”; “prevention bundle”; “care checklist”; “patient care bundles”; “clinical risk”; “risk management”.

### 2.3. Inclusion and Exclusion Criteria

Studies were included if they described the steps for the development of BCB in the management of clinical risk situations in patients in a care pathway in any setting. The eligibility criteria for studies (Box 1) are described according to the PICOS framework (population or problem, intervention, comparison, outcome, and type of study). All studies that did not describe the development process were excluded [Appendix A]. A data extraction form was prepared to report the characteristics of the publications, including authors’ surname, database, year of publication, country, study design, type of Care bundle, presence of Care bundle phases, and presence of the care bundle development process.

Box 1Inclusion criteria for this review.**Population or problem** = Patients in a clinical/care pathway**Intervention** = Care Bundle/s**Comparison** = Not applicable**Outcome** = Describe the process of developing the care bundle in clinical risk situations in a care pathway in any context**Study type** = All primary studies and systematic reviews in English, Spanish, and Italian from 2005 to 2023

### 2.4. Data Extraction and Study Quality

The search results were managed using the bibliographic reference software Zotero. The data were entered into a Microsoft Excel^®^ file via a table structure to enable information management. To enable independent work, the reviewers shared their work through Rayyan and Google Drive^®^ and through meetings on the Microsoft Teams^®^ platform.

Duplicate articles were removed, and the titles and abstracts of all collected papers were reviewed by E.S., M.S., and M.C. who conducted study selection (screening, eligibility, and inclusion) during each review phase. The titles, abstracts, and full texts were screened and thoroughly analyzed for correspondence with the inclusion criteria. Three reviewers confirmed eligibility based on the full text of the relevant articles, and in the case of disagreement, a consensus was reached through discussion. Data extraction was conducted by E.S., M.S., and M.C. and a data extraction form was formulated by assessing the characteristics of the publications.

The methodological quality of the studies was assessed by two reviewers, E.S. and M.S., using the Joanna Briggs Institute’s *JBI Critical Appraisal Tool* with a checklist appropriate to each type of study included. A narrative summary was developed to specify the BCB development method used.

### 2.5. Data Synthesis

Two reviewers (E.S. and M.S.) performed a double data entry. The characteristics of the reviews were summarized using frequencies and percentages for categorical variables. We then recorded all components related to the strategies used to develop the proposed care bundles in the included studies, assessing the frequency of each.

## 3. Results

### Overview

The search strategy identified a total of 1372 potentially relevant articles. After removing duplicates and reading the abstracts, 243 full texts were evaluated and of these, 16 articles were included in the review (Figure 1).

All of the included studies (Table 1) dealt with the BCB development process. A total of 11 articles were conducted in the United States [1,12,13,14,15,16,17,18,19,20]: 4 in Europe [21,22,23,24] and 1 in Australia [25]. With regard to the study types, two are intervention studies [21,25], three are quality improvement [12,13,16], one study is a randomized controlled trial [13], three studies are observational studies [15,23,25], one study is a systematic review [24], two studies are cohort studies [17,18], three studies are expert opinion [1,19,21], and one record is a mixed method study [20].

Among the risk situations, the 16 articles dealt with 5 articles [12,14,17,18,25] about bundles for the prevention of surgical wound infection, Surgical Site Infection (SSI), 2 articles [21,22] about bundles for the prevention of Ventilator-Associated Pneumonia (VAP) ventilator-associated infections, 2 articles [1,19] about bundles for the prevention of Central Venous Catheter Central Line-Associated Bloodstream Infection (CLABSI)-related infections, 1 article [16] about the bundles for the prevention of joint replacement associated infections, 2 articles [1,24] dealt with bundles for the prevention of both ventilator-associated infections (VAP) and central venous catheter-related infections (CLABSI), 2 articles [13,26] dealt with bundles for the prevention of Bladder Catheter-associated Urinary Tract Infection (CAUTI), 2 articles [15,23] dealt with bundles on the prevention of sepsis (SEPSIS), and 1 article [20] with bundles in the management of transitions of care [Appendix A].

Acun et al. [22] described the bundle development process using the CDC guidelines and those of the General Directorate of Public Health affiliated with the Ministry of Health of the Republic of Turkey. A literature review was conducted, and expert opinions (an infectious disease physician and four experienced infection control nurses) were gathered.

Andiman et al. [12] detailed their bundle development process, which involved forming a multidisciplinary expert group. After a literature review to identify the best evidence for preventing surgical site infections, they identified the steps to include in the bundle, drawing from colorectal surgery and other specialties.

Andreessen et al. [13] based their bundle development process on CDC criteria and guidelines. A multidisciplinary team was formed, including two infection control nurses, two education nurses, three clinical nurses, two surgical nurses, four nurse leaders (operating room, emergency room, intensive care unit, and medical wards), two clinical nurse leaders, one urologist, two computer scientists, and one primary caregiver.

Anthony et al. [14] included a literature review of current evidence in their bundle development process.

Baldwin et al. [23] used the Surviving Sepsis Campaign guidelines for the management of severe sepsis and septic shock, followed by the Delphi method to reach a consensus.

Borgert et al. [24] developed a flowchart for Central Line-Associated Bloodstream Infection (CLABSI) and Ventilator-Associated Pneumonia (VAP).

Bruce et al. [15] used the Intensive Care Society Guidelines in their bundle development process, and subsequently, a multidisciplinary healthcare team implemented the 2008 and 2012 SSC guidelines, adapting them to the hospital protocol.

Bullock et al. [16] conducted a literature review on surgical wound infections following primary total joint arthroplasty surgery and then formed a multidisciplinary team to formulate the care bundle, including three surgeons specializing in total joint arthroplasty (TJA), anesthetists, infectious disease specialists, nurses, physiotherapists, and coordinators.

Concerning the two articles by Davidson et al. [17,18], the authors assessed the impact of the bundle on preventing surgical wound infections from two different perspectives: one on reducing surgical wound infections after hysterectomy and the other on reducing surgical site infection rates after cesarean section. In both articles, the source for the bundle development was the CDC guidelines. In the article on reducing surgical site infection rates after cesarean section, the results of a literature review on colorectal surgery, demonstrating the reduction in superficial surgical site infection rates, were also integrated.

Dieplinger et al. [25] used the Comprehensive Unit-based Safety Program (CUSP) for the development process. The starting point was identifying a set of evidence-based interventions from the literature, followed by using the CUSP approach to incorporate them into the bundle phases.

Giles et al. [26] described a development process involving a literature review phase, using evidence-based material (guidelines, protocols regarding the use of the permanent urinary catheter), and finally consulting with ward staff on implementation strategies.

Muller et al. [19] began their development process using HICPAC guidance documents and other published documents compiled in a literature review, followed by a consensus among pediatric experts (including pediatric infectious disease specialists), neonatologists, and infection control nurses.

Resar et al. [1] specified the presence of an experienced team to decide on the bundle components/phases as the only aspect of the development process.

Rosgen et al. [20] explicitly mentioned a consensus meeting among stakeholders as the only element of the development process.

Speck et al. [21] conducted a literature review and subsequently used a modified two-stage Delphi method, resulting in the identification of 19 interventions, 5 of which were process-related and 14 structural.

## 4. Discussion

### 4.1. Principal Findings

This systematic literature review identifies the development process of Care Bundles (BCB) in the management of clinical risk situations within care pathways in any context.

Despite the substantial literature published on the subject, few studies have provided a detailed description of the care bundle design process. The strategies or steps described in the studies were identifying an expert working group; identifying a critical care theme (proposal of the care bundle); defining a cluster of interventions within the theme; conducting a systematic literature search in each of these areas; evidence synthesis; assessment of strengths and limitations of the evidence; consensus method process; and implementation of the care bundle in a pilot study.

By specifically analyzing the articles, it can be established that 4 articles benefited from CDC guideline criteria: Acun et al. [22], Andiman et al. [12], Davidson et al. [17,18]; 4 used the Delphi method: Baldwin et al. [23], Borgert et al. [24], Bullock et al. [16], Speck et al. [21]; 7 articles utilized literature reviews: Anthony et al. [14], Borgert et al. [24], Bullock et al. [16], Davidson et al. [17], Giles et al. [26], Muller et al. [19], Speck et al. [21]; and 11 used a multidisciplinary team to include or exclude certain activities in the BCB: Acun et al. [22], Andiman et al. [12], Andreessen [13], Borgert et al. [24], Bruce et al. [15], Bullock et al. [16], Dieplinger et al. [25], Giles et al. [26], Muller et al. [19], Resar et al. [1], Speck et al. [21]. Only one article by Borgert et al. [24] described, in detail, a flowchart for the development of BCB with seven steps: identify the problems/risks in a specific patient population or intervention that contribute to serious harm and/or high costs; define the identified care problems or risks; conduct a literature search to gather relevant evidence on the problems/risks and to find related items; select potentially relevant and feasible items from the search; select a final set of up to five items; create the care bundle in draft form; and finally, test the care bundle with a pilot study to assess its reliability.

The BCBs most featured in clinical risk management in this review are prevention of ventilator-associated pneumonia (VAP) [21,22,24], prevention of central venous catheter-related bloodstream infections (CLABSIs) [1,19,24], prevention of surgical site infections (SSIs) [12,14,17,18,25], prevention of catheter-associated urinary tract infections (CAUTIs) [13,26], and finally, prevention of sepsis [15,23]. These are the most common BCBs, although in recent years, this tool has also been used for other specific clinical situations such as falls, pressure injuries, emergency situations, and transitions of care.

Situations were identified in which clinical risk management can be supported by specific BCBs, which are VAP, CLABSI, CAUTI, SEPSIS, and SSI. These results may provide guidance for the development and implementation process of BCBs to manage clinical risk situations in care settings.

It should be noted that, given the heterogeneity of study designs, their descriptive nature, and the lack of a standard methodology, it is difficult to objectively describe the contents of the articles, which was confirmed after assessing quality using the JBI Critical Appraisal Tool.

### 4.2. Limitations

“Among the limitations of the conducted review, we found that almost all the identified articles demonstrated the effectiveness of BCB without making the development process explicit. An additional limitation of the review was that it did not consider the presence of tools in the literature to assess the quality of BCB. Due to the previously mentioned limitations, the systematic review acknowledges the need for more comprehensive evidence to identify a process or methodology for BCB development, while at the same time providing a unique synthesis of trends and gaps in literature”.

## 5. Conclusions

This study has identified the development processes of Care Bundles (BCBs) in clinical risk scenarios. BCBs are useful and clear tools that facilitate compliance monitoring by all members of the healthcare team.

The review highlighted effective methods for designing evidence-based care packages.

However, the limited number of studies addressing the methodology for developing BCBs is one of the main limitations of this review.

These limitations encourage further research, including a thorough and rigorous analysis of BCBs in specific care contexts; validating the flowchart identified in this review through a pilot study; and conducting a review to assess the availability of useful tools for evaluating the quality of BCBs that can guide professionals in the care package design process.

In Italy, there is growing interest in using care packages in various clinical contexts, as they provide professionals with a practical method to implement evidence-based practice. It is essential to encourage further research that goes beyond descriptive studies.

In summary, this work contributes to clarifying the development process of BCBs and underscores the need for further investigations to optimize the effectiveness of these intervention strategies.

## Figures and Tables

**Figure 1 healthcare-12-02242-f001:**
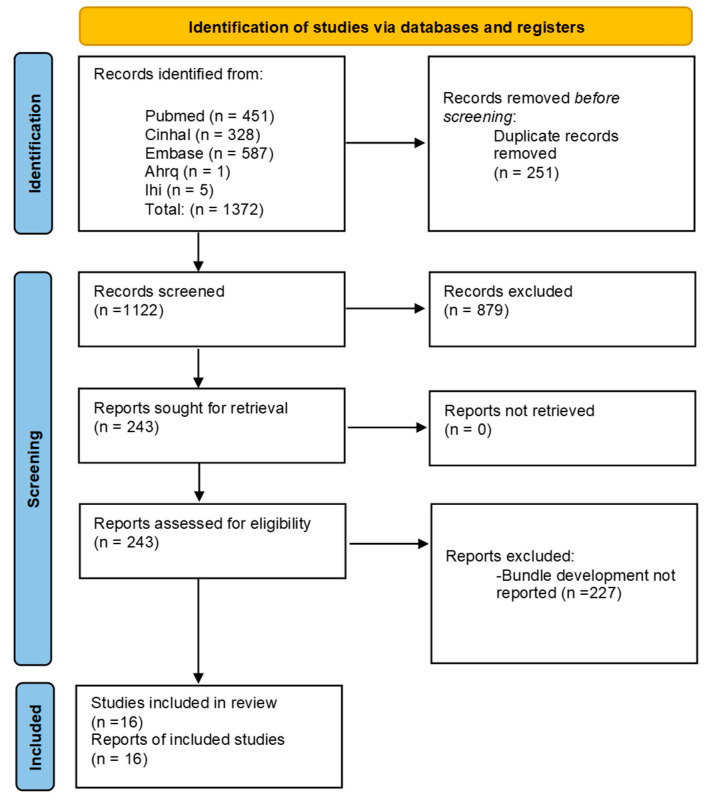
PRISMA (Preferred Reporting Items for Systematic Reviews and Meta-Analyses) flow diagram of the included studies.

**Table 1 healthcare-12-02242-t001:** Characteristics of the included studies.

First Author (Year of Publication)	Title	Country	Situation Risk	Method of Developing
Acun A (2022) [22].	Assessing the Efficacy of Ventilator-Associated Event Prevention bundles in the Intensive Care Units: An Intervention Study	Turkey	VAP ^1^	CDC ^2^ criteria and guidelines, Guidelines prepared by the General Directorate of Public Health affiliated to Republic of Turkey Ministry of Health Literature review Opinions of one infection control doctor and four infection control nurses, who were experts in their fields
Andiman SE (2018) [12].	Decreased Surgical Site Infection Rate in Hysterectomy: Effect of a Gynecology-Specific bundle	USA ^3^	SSI ^4^	The bundle was designed and implemented by the committee led by representatives from gynaecology, anesthesia, hospital epidemiology and infection control, and peri- and postoperative nursing leadership The group developed a gynecology-specific bundle based on existing evidence and best practice guidelines, some of which were extrapolated from colorectal and other surgery literature
Andreessen L (2012) [13].	Preventing catheter-associated urinary tract infections in acute care: the bundle approach	USA	CAUTI ^5^	CDC criteria and guidelines A multidisciplinary team was assembled. The team included 2 infection control nurses, 2 nurse educators, 3 staff nurses, 2 surgical nurses, 4 nurse managers (operating room, ER, ICU, and medical units), 2 clinical nurse leaders, 1 urologist, 2 information technologists, and 1 chief medical resident
Anthony T (2011) [14].	Evaluating an evidence-based bundle for preventing surgical site infection: a randomized trial.	USA	SSI	A review of the literature was conducted that focused on evidence-based measures that could be expected to reduce SSI and could be instituted in our practice setting
Baldwin LN (2008) [23].	An audit of compliance with the sepsis resuscitation care bundle in patients admitted to A&E with severe sepsis or septic shock	United Kingdom	SEPSIS ^6^	Surviving Sepsis campaign guidelines for the management of severe sepsis and septic shock. Development of the package with consensus (Delphi consensus method)
Borgert M (2017) [24].	A flowchart for building evidence-based care bundles in intensive care: based on a systematic review	Netherlands	VAP; CLABSI ^7^	Identify problems or risks in a specific patient population or intervention that contributes to great harm and/or high cost (Systematic Reviews-Adverse Event Trigger Tool) The identified care problems or risks should be clearly defined (comprehensive literature search strategy) Conduct a literature search to collect relevant evidence for the problems or risks and to find related elements (collect evidence from the international electronic databases and from the distillation from inter-national clinical guidelines Select potential relevant and feasible elements from the literature search (select those elements that were described in the literature and were associated with the identified problem or from local or (inter)national clinical guidelines) Select a final maximal set of five elements (GRADE approach to evaluate the quality of the evidence of the elements) (weighing and scoring technique to select the most suitable, reliable, or most appropriate key elements root cause analyses; FMEA; through discussion sessions or consensus meetings with experts or hospital staff) Create the care bundle in draft form (create the bundle in draft form and check if the IHI ^8^ bundle requirements are met) Pilots test the care bundle in order to assess the reliability (the pilot should be performed in a small sample of patients to identify (potential) risks or barriers for implementation). It is important to monitor the performance of all bundle elements to identify potential problems or risks and to evaluate if the care bundle is feasible, comprehensive, effective, and easy to use)
Bruce HR (2015) [15].	Impact of nurse-initiated ed sepsis protocol on compliance with sepsis bundles, time to initial antibiotic administration, and in-hospital mortality	USA	SEPSIS	SSC guidelines A multidisciplinary health care team to implement the 2008 and 2012 SSC guidelines Adaptation of the hospital protocol
Bullock MW (2017) [16].	A bundle Protocol to Reduce the Incidence of Periprosthetic Joint Infections After Total Joint Arthroplasty: A Single-Centre Experience	USA	INFECTIONS	In 2011, our institution conducted a comprehensive review of primary TJA cases Creation of a multidisciplinary team focused on formulating a “bundle” to optimise patient outcomes. Our team included 3 fellowship-trained TJA surgeons, anesthesiologists, infectious disease specialists, nurses, physical therapists, and administrative coordinators
Davidson C (2020) [17].	Impact of a surgical site infection bundle on caesarean delivery infection rates	USA	SSI	Using CDC guidelines Integration into the CDC guidelines of the results of a review of the literature on colorectal surgery in which the reduction in superficial SSI rates was demonstrated
Davidson C (2020) [18].	Reducing abdominal hysterectomy surgical site infections: A multidisciplinary quality initiative	USA	SSI	3. CDC criteria and guidelines
Dieplinger B (2020) [25].	Implementation of a comprehensive unit-based safety programme to reduce surgical site infections in caesarean delivery	Austria	SSI	Initiation of a Comprehensive Unit-based Safety Program (CUSP) We introduced a bundle of evidence-based interventions We implemented the evidence-based bundle using the CUSP approach into clinical routine
Giles M (2015) [26].	Does our bundle stack up! Innovative nurse-led changes for preventing catheter-associated urinary tract infection (CAUTI)	Australia	CAUTI	Exploration of the literature Collaboration with all stakeholders and development of evidence-based IUC insertion criteria, care bundles, and guidelines for the nurse-led protocols Further consultation with ward staff related to implementation strategies, nomination of ward champions to engage ward staff and assist in implementation of the nurse-led protocol
Muller M (2023) [19].	Neonatal Intensive Care Unit (NICU) White Paper Series: Practical approaches for the prevention of central-line-associated bloodstream infections	USA	CLABSI	Practical approaches in a question-answer format, with responses based on the consensus opinion of pediatric experts, including pediatric infectious disease specialists, neonatologists, advanced practice nurse practitioners, and infection experts, convened by SHEA using the guidelines as documents HICPAC guidance and other published documents collected from a literature review
Resar R (2012) [1].	Using Care Bundles to Improve Health Care Quality	USA	VAP; CLABSI	Team members from the ICUs participating in the collaborative community Definition of the bundle phases
Rosgen BK (2022) [20].	Co-development of a transitions in care bundle for patient transitions from the intensive care unit: a mixed-methods analysis of a stakeholder consensus meeting.	Canada	TRANSITIONS	A stakeholder consensus meeting
Speck K. (2016) [21].	A systematic approach for developing a ventilator-associated pneumonia prevention bundle	USA	VAP	Identification of potential interventions to include through a review of current guidelines and the literature Implemented a 2-step modified Delphi method to obtain consensus on the final list of interventions An interdisciplinary group of clinical experts participated in the Delphi process, led by a group of technical experts Identified 65 possible interventions. Through the Delphi method, the list was narrowed down to 19 interventions that included 5 process measures and 14 structural measures

^1^ VAP: Bundle for the prevention of pneumonia associated with mechanical ventilation. ^2^ CDC: Centers for Disease Control and Prevention. ^3^ USA: United States of America. ^4^ SSI: Bundle for the prevention of surgical site infections. ^5^ CAUTI: Catheter-associated urinary tract infection prevention bundle. ^6^ SEPSIS: Sepsis prevention bundle. ^7^ CLABSI: Bundle for the prevention of central line-associated bloodstream infections. ^8^ IHI: Institute for Healthcare Improvement.

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
