# Peer review of "Identifying the Bundle/Care Development Process in Clinical Risk Management: A Systematic Review"

_healthcare, 2024, doi:10.3390/healthcare12222242_

Round 1
Reviewer 1 Report
Comments and Suggestions for Authors
I ve read with interest the paper. The aim and scope are coherent to those proposed by the journal. However I have some concerns:
1) Abstract: the abstract is too long. I suggest to modify it in a shorter way. Particularly, the section pf the results is too long.
2) Background: I suggest to avoid paragraphs sometimes it is redundant. So I suggest to collate some of this paragraphs.
3) Methods are well described. I just suggest to avoid form such as "we ..."
4) Results are well described.
5) discussion I think that conclusions should be in a separate section.
6) references are not in the journal style.
Author Response
Grazie per l'attento esame del mio manoscritto. Ho preso in considerazione tutti i vostri commenti e suggerimenti e ho apportato le seguenti modifiche, che troverete nel file Word allegato per una facile consultazione. Rimango a disposizione per ulteriori chiarimenti e attendo con ansia il vostro feedback.
Grazie ancora per il vostro prezioso contributo.
Cordiali saluti,Dott
. Emanuele Sebastiani

Reviewer 2 Report
Comments and Suggestions for Authors
Dear Authors,
Congratulations on your excellent work. I hope the following suggestions will help to improve your paper further.
First, I suggest consolidating the introduction into a cohesive paragraph rather than dividing it into subparagraphs with question/answer formats.
Additionally, several statements need citations; for example, no references are provided in section 1.1.
Section 1.3 should be revised entirely, with a more deductive approach to presenting the application contexts of BCBs.
Section 1.4's title must be clarified and function well as a heading. For this reason, I recommend integrating this section into a single introduction paragraph.
Moreover, the introduction should provide a more thorough exploration of the research gaps and objectives. It would also be helpful to clarify why the period 2005-2023 was selected—was there a specific reason? You previously mentioned the development of the concept in 2001. Finally, I suggest addressing a highly relevant issue in clinical risk management, where BCBs could play a critical preventive role, such as falls in hospital settings (please refer to DOI: https://doi.org/10.1177/25160435241246344). Additionally, it would be beneficial to elaborate on the mechanisms of clinical risk management in the introduction.
Regarding the methodology, please clarify why Scopus was not considered for consultation. Please specify.
In text box 1, details such as the language of the articles and the study period are missing, which might not be essential for publication. Did you use a reference management tool such as Endnote?
In Table 1, only 15 out of the 16 included papers are listed. Could you clarify this discrepancy?
Given the number of articles included, the discussion section appears rather brief. I encourage you to elaborate on the results, providing more detailed and scientifically meaningful insights into the research gap you aim to address.
Lastly, the conclusions appear redundant. Please consider summarizing them to make the message more concise and impactful.
I also recommend revising the abstract to make it more concise and engaging, as much of the current information can be streamlined.
After these changes, the manuscript will be ready for publication.
Thank you
Author Response

(The authors gave the same response as above.)

Reviewer 3 Report
Comments and Suggestions for Authors
This manuscript describes a systematic review of the literature regarding processes to develop Bundles/Care Bundles. While BCBs have great value in clinical applications to improve care and reduced mortality and morbidity, it would seem there is sparse information regarding their development.
While the aim of the study to better understand BCB development may have value, it would seem there is insufficient and inconsistent evidence to provide true value from this work. Out of 1372 papers found in the literature, there were only 16 that described the BCB process! Even in these, the process varied quite a bit and really reduced to determination of BCBs by discussion of multi-disciplinary experts (11 of 16 papers). If this is the primary conclusion of the work, it is unclear how this will move the field forward and what value it provides to extend knowledge in this area. Therefore, this works seems to have limited impact overall.
The work is very descriptive in nature. It would benefit from more rigorous analysis.
Author Response

(The authors gave the same response as above.)

Round 2
Reviewer 2 Report
Comments and Suggestions for Authors
Dear Authors,
I am pleased to inform you that your manuscript has undergone substantial improvements and is now ready for publication.
Best regards
Reviewer 3 Report
Comments and Suggestions for Authors
It seems much of the revision centers around a reworked "Introduction" review and minimally revised "Conclusion", as well as the abstract. The remainder of the manuscript which influenced the prior review is essentially unchanged. As such, many of the original concerns remain, namely, that this systematic review does not add much value to extend the knowledge of BCB development and usefulness in clinical practice. The work has limited rigor and is extremely descriptive in nature.
The response and revision does not ally the concern that there remains only 16 articles in the review that the authors attempted to glean information about BCB development and to make recommendations. Due to the dispersion of the content in these articles, the findings are anecdoctal at best and seems insufficient to warrant publication.
It is unclear how publication of this manuscript advances the field.
